

# Cloning, phylogenetic research, and prokaryotic expression study of the metabolic detoxification gene *EoGSTs1* in *Empoasca onukii* Matsuda

Yujie Zhang[1,2], Wenlong Chen[1,2], Ming Li[3], Lin Yang[1,2] and Xiangsheng Chen[1,2]

[1] Institute of Entomology, Guizhou University, Guiyang, China
[2] Guizhou Key Laboratory for Plant Pest Management of Mountainous Region, Guizhou University, Guiyang, China
[3] The Provincial Special Key Laboratory for Development and Utilization of Insect Resources of Guizhou, Guiyang, China

## ABSTRACT

Due to the misuse of chemical pesticides, small green leafhoppers (*Empoasca onukii* Matsuda) have developed resistance to pesticides, thereby posing a serious problem to the tea industry. Glutathione S-transferases (GSTs) are an important family of enzymes that are involved in pesticide resistance in *Empoasca onukii* Matsuda. *Empoasca onukii* GST sigma 1 (*EoGSTs1*, GenBank: MK443501) is a member of the GST family. In this study, the full-length cDNA of *EoGSTs1* was cloned by reverse transcription polymerase chain reaction (qPCR), and its taxonomic identity was examined. Furthermore, we performed bioinformatics and phylogenetic analyses of the gene and structural and functional domain prediction of the protein. The results demonstrate that EoGSTS1 belongs to the Sigma family of GSTs; the full-length *EoGSTs1* cDNA is 841 bp with a 624-bp coding region that encodes a 23.68932-kDa protein containing 207 amino acids. The theoretical isoelectric point (IEP) was calculated to be 6.00. Phylogenetic analysis indicates that EoGSTS1 is closely related to the *Sub psaltriayangi* subfamily of the *Cicadoidea* superfamily in order *Hemiptera*, whereas it is distantly related to *Periplaneta americana* of order *Blattodea*. Amino acid sequence alignment of EoGSTS1 and GSTs from four other insects of order *Hemiptera* revealed protein sequence conservation. Tertiary structure analysis and structural domain functional predictions of the protein revealed that EoGSTS1 contains nine $\alpha$ helices and two $\beta$ sheets with one conserved GST domain. The results of enzyme activity assay showed that recombinant *EoGSTs1* (rEoGSTs1) protein had catalytic activity for substrate 1-chloro-2,4-dinitrobenzene (CDNB) and exhibited the highest activity at pH 7 and 25 °C. The Michaelis constant Km of rEoGSTs1 protein was $0.07782 \pm 0.01990$ mmol/L, and the maximum reaction rate Vmax was $12.15 \pm 1.673$ µmol/min·mg. Our study clarified the taxonomic identity of small green leafhopper *EoGSTs1* and revealed some properties of the gene and its encoded protein sequence. According to the catalytic activity of the rEoGSTs1 enzyme on the model substrate CDNB, we infer that it functions in the degradation of exogenous substances.

Corresponding author
Xiangsheng Chen,
chenxs3218@163.com,
xschen@gzu.edu.cn

## INTRODUCTION

Small green leafhoppers or tea green leafhoppers, *Empoasca onukii* Matsuda, belong to the *Cicadidae* family in genus *Jacobiasca formosana*. Small green leafhoppers are one of the most predominant insects that infiltrate tea production fields. Small green leafhoppers are extremely harmful to tea trees, causing a 15%–50% loss of production of tea each year in mainland China and Taiwan (*Niu et al., 2015*; *Hazarika, Bhuyan & Hazarika, 2009*; *Fu, Han & Xiao, 2014*; *Saha & Mukhopadhyay, 2013*; *Qin et al., 2015*; *Wang, 2004*; *Zhang & Chen, 2015*; *Mu et al., 2012*). To date, the pest control and prevention of small green leafhoppers still mainly rely on chemicals. Due to long-term misuse and abuse of chemical compounds, insects have developed resistance to most chemical pesticides (*Zhan et al., 2012*). For instance, *Wang (2004)* performed drug membrane interaction tests on the resistance of small green leafhoppers to 14 chemical pesticides and found that their LC50 values and resistance gradually increased with the higher pesticide doses used in tea tree facilities. In particular, the resistance to thiamethoxam increased by 2.2- to 13.3-fold, which is the largest increase among all the chemical pesticides.

Glutathione S-transferases (GSTs) are a family of enzymes that widely exist in aerobic organisms and play an important role in detoxifying insects from endogenous or exogenous toxins (*Enayati, Ranson & Hemingway, 2005*). For example, *Wu et al. (2016)* used cycloxaprid as a toxin to induce the activities of detoxification enzymes in Alfalfa aphid and found that the activities of GSTs in this species that survived treatment were significantly higher than those in control insects. Thus, they suggested that GSTs play a critical role in detoxifying cycloxaprid in Alfalfa aphid. *Zhang et al. (2014)* studied the levels of pest resistance to the neonicotinoid class of insecticides and the synergistic effect of enzyme inhibitors in *Aphis gossypii* in different regions of Shandong Province, China. They discovered a noticeable synergistic effect on this class of insecticides by the enzyme inhibitor diethyl maleate (DEM). Cytoplasmic GSTs in insects are divided into six subfamilies, which include Epsilon, Delta, Zeta, Omega, Sigma, and Theta (*Chelvanayagam, Parker & Board, 2001*). To date, studies on resistance to pesticides related to GSTs in insects have focused on the Epsilon and Delta subfamilies, whereas investigations on the involvement of other subfamilies of GSTs are limited (*Yu et al., 2010*). In recent years, with the rapid development of techniques in molecular biotechnology, studies on other GST subfamilies related to pesticide resistance have grown in number, although studies on GST-related pesticide resistance in *E. onukii* Matsuda remain scarce.

Pesticide resistance in *E. onukii* can further increase in the foreseeable future because of current continuous misuse of chemical pesticides. Reducing or slowing down this process of increasing pesticide resistance has thus become a major goal. Research on genes involved in the resistance of small green leafhoppers can provide critical and fundamental knowledge for the development of new pesticides, recipes for pesticide cocktails, or

protocols for alternating the usage of different chemicals (*Zhou & Han, 2017*). GSTs are the major enzymes involved in metabolic resistance and have been proven essential in the detoxification of small green leafhoppers that are resistant to chlorpyrifos and thiamethoxam. In this study, we screened transcriptome data and identified genes that are differentially expressed in *E. onukii* upon chemical treatment and discovered that the expression level of the GST *cluster-166.0* was significantly upregulated. Therefore, we cloned GST *cluster-166.0* using reverse transcription polymerase chain reaction (qPCR) with proofreading. Bioinformatics analysis, phylogenetic reconstruction, and homology modeling of the full-length cDNA and its encoded protein sequence were conducted to obtain biological information on the *E. onukii* GST gene (*EoGSTs1, cluster-166.0*), including its physical and chemical properties, taxonomic classification, and the spatial structure of the EoGSTS1 protein. In addition, the expression vector of the EoGSTS1 protein was constructed for the expression in prokaryotic cells, and the expression of fusion protein was detected by western blotting. Finally, the enzyme activity of purified EoGSTS1 protein was determined, and its enzymatic characteristics were verified. This information may serve as the foundation for future applications of *EoGSTs1*, may be used as a theoretical reference for the detection and comprehensive management of chemical resistance in small green leafhoppers, as well as employed in the development and utilization of new pesticides.

## MATERIALS AND METHODS

### Preparation of insect samples

*E. onukii* Matsuda used in this study is a sensitive strain that has been cultured in the laboratory for many generations. Insects were maintained in a light incubator with a constant temperature of 25 °C ± 1 °C, a humidity of 70% ± 5%, and a photoperiod of 16:8 (L:D). Small green leafhoppers of approximately the same size were starved for 1 h and then treated with 2.5 µg/mL of thiamethoxam for 48 h. Individuals who survived the treatment were collected at 20 insects/tube, flash frozen in liquid nitrogen, and stored in a −80 °C freezer.

### Extraction and examination of total RNA

Total RNA was extracted using the TRIzol® PlusRNA Purification Kit (Cat. No. 12183-555, Invitrogen, Carlsbad, CA, USA) following the manufacturer's instructions. The extracted RNAs were examined by NanoDrop quantification and electrophoresis. Three microliters of each extracted RNA sample were used in agarose gel (1%) electrophoresis.

### Primer design and template synthesis

Primers were designed using Primer Premier 6.0 (Premier Biosoft International, Palo Alto, CA, USA). Specific primer sequences are listed in Table 1. The 5′ rapid amplification of cDNA ends (RACE) and 3′ RACE templates were synthesized using a GeneRacer™ Kit (Cat. No. 12183-555, Invitrogen, Carlsbad, CA, USA). The cDNA template used for intermediate fragment PCR was synthesized using the SuperScript™ III First-Strand Synthesis SuperMix (Cat. No. 18080-400, Invitrogen, Carlsbad, CA, USA) for qPCR Kit.

*PCR*: The reaction system and conditions for 5′ RACE PCR, intermediate fragment PCR, and 3′ RACE PCR are presented in Tables S1–S10.

**Table 1 Primers and sequences.**

| Primer | Sequence (5′ to 3′) |
| --- | --- |
| rGST-R1 | CTCCTGCCTTACCCTCGATCTCAAGAAT |
| rGST-R2 | CAGGCCATTCACCCCATTTGACTCTG |
| mGST-F | GGGCTTGGAGAACCCATAAGATT |
| mGST-R | GATATCTGACCCGTTCATGTTGC |
| rGST-F1 | GAGAGGGGAGTTGGCGAAATACTACTACGA |
| rGST-F2 | GGCGAAATACTACTACGAGCGTGATGAAGAG |

## Prokaryotic expression and purification

Full-length splicing primers were designed based on PCR-based accurate synthesis (PAS). Protective bases were designed at both ends of the primers to synthesize the *EoGSTs1*-TARGET gene, which connected the site between the *Nde* I and *Xba* I sites of vector PCZN1. The obtained recombinant plasmid PCZN1-*EoGSTs1*-TARGET was transferred into *Escherichia coli* strain Top10, and positive clones were selected for sequencing and restriction enzyme digestion to ensure the correct expression plasmid was obtained through subcloning.

Approximately one μL of the constructed expression plasmid was added to 100 μL of Arctic Express *E. coli*, placed on the ice for 20 min, heat-shocked for 90 s at 42 °C, placed on ice for 5 min, added 600 mL of LB culture medium, and placed on a shaker for 1 h at the frequency of 220 rpm and 37 °C. After centrifugation, all the samples were coated onto LB plates that contained 50 μg/mL Amp (ampicillin), and incubated overnight at 37 °C. Monoclones were selected and inoculated into three mL of LB medium containing 50 μL/mL, placed on a shaker overnight at the frequency of 220 rpm and 37 °C, inoculated into 30 mL of LB medium (containing 50 μL/mL AMP) at a ratio of 1:100 the next day, and placed on a shaker at a frequency of 220 rpm and 37 °C when the OD600 value of the culture reached 0.6–0.8. Then, one mL of the culture was collected and centrifuged for 2 min at 10,000 rpm at room temperature. The supernatant discarded, and the pellet was resuspended in 100 μL 1× sample loading buffer. IPTG was added to the remaining culture until the final concentration was 0.5 mM and placed on a shaker overnight at a frequency of 220 rpm and 11 °C to induce the expression of the fusion protein. Then, one mL of the culture was collected and centrifuged for 2 min at 10,000 rpm at room temperature, the supernatant was discarded, and the pellet was resuspended in 100 μL of 1× sample loading buffer. The remaining culture was centrifuged for 10 min at 4,000 rpm, the supernatant was discarded, and the pellet was resuspended in phosphate-buffered saline (PBS). The cells in suspension were then lysed by sonication, the supernatant and pellet were resuspended in sample loading buffer, detected by 12% SDS-PAGE, and stained with Coomassie brilliant blue.

The results of the preliminary experiment showed that the target protein was soluble. Therefore, the supernatant was purified by Ni affinity chromatography. The supernatant solution (flow rate: 0.5 mL/min) was loaded onto the Ni-IDA-Sepharose Cl-6B affinity

chromatography column (pre-equilibrated with Ni-IDA binding buffer) with a low-pressure chromatography system and washed with Ni-IDA binding buffer at the flow rate of 0.5 mL/min until the OD280 value of the effluent reached the baseline. The column was washed with Ni-IDA washing buffer (20 mM Tris–HCl, 20 mM imidazole, 0.15 M NaCl, pH 8.0) at the flow rate of one mL/min until the OD280 value of the effluent reached the baseline. Then, the target protein was eluted with Ni-IDA elution buffer (20 mM Tris–HCl, 250 mM imidazole, 0.15 M NaCl, pH 8.0) at a flow rate of one mL/min, and effluent was collected. The above-mentioned protein solution was poured into a dialysis bag, and PBS (3.58 g $Na_2HPO_4 \cdot 12H_2O$, 8.8 g NaCl, 0.2 g KCl, 0.27 g $KH_2P0_4$) was used for overnight dialysis, then detected by 12% SDS-PAGE.

## Western blot analysis

A three-$\mu$L sample was loaded into a well of a polyacrylamide gel. After running the stacking gel at 90 V and then increased to 200 V until the end of electrophoresis, conducted PVDF transfer (constant pressure 100 V, 1.5 h, constant current 250 mA), the membrane was washed four times (5 min each time) with phosphate-buffered saline-Tween 20 (PBST) and then kept in the blocking buffer containing 5% non-fat milk for 1 h at 37 °C. The membrane was incubated overnight with a term primary antibody that was diluted in the blocking buffer (dilution rate, 1:1000). The next day, the membrane was washed four times (5 min each time) with PBST. The membrane was then incubated with a secondary HRP antibody that was diluted with the blocking buffer with 5% milk (dilution rate, 1:5000) for 1 h at 37 °C. Then, the membrane was washed four times (5 min each time) in a clean box, developed, and exposed by the ECL (electrochemiluminescence) method.

## Enzyme activity determination

Parameters of enzymatic kinetic: The determination method was based on *Li et al. (2018a)*; *Li et al. (2018b)* with minor modifications. The reaction system consisted of phosphate buffer (100 mmol/L), rEoGSTs1 protein (0.12 mg), and GSH (0.01, 0.02, 0.04, 0.06, 0.08, and 0.1 mmol/L). After mixing the above solution, the mixture was incubated with 1-chloro-2,4-dinitrobenzene (CDNB) solution (10 mmol/L) at 35 °C for 15 min, then 20 $\mu$L of the CDNB solution was added into the mixture, and the volume was set to 200 $\mu$L. Subsequently, the absorbance was measured at a wavelength of 340 nm in the Multiskan GO 1510 spectrophotometer (Thermo Fisher Scientific, Vantaa, Finland), 30 s for each time and for a total of 2 min. Enzyme activity was calculated using the recombinant-inactivating protein as control. The specific activity was calculated according to the following formula, and the obtained data were submitted to GraphPad Prism 8.0.1 to calculate for enzyme kinetics using the following equation:

$$\text{Specific activity (}\mu\text{mol/min·mg)} = \frac{\Delta OD340 \times V}{\varepsilon \times T \times L \times E}.$$

Note: OD340: Change in absorbance per unit time; V: enzymatic reaction volume (200 mL); $\varepsilon$: extinction coefficient of product (9.6 mmol/cm); T: reaction time (2 min); L: optical path (one cm); and E: amount of enzyme added (0.12 mg).

To determine the effect of temperature on the activity of the purified rEoGSTs1 protein, Glutathione S-transferase (GSH-ST) assay kit (Colorimetric method) was used (Cat. No.

A004, Nanjing Jiancheng Bioengineering Institute, Nanjing, Jiangsu, China), following the manufacturer's protocol. Approximately 0.12 mg of the rEoGSTs1 protein was added into the reaction system and then was incubated at 5 °C, 15 °C, 25 °C, 35 °C, 45 °C, and 60 °C for 15 min.

To determine the effect of pH on the activity of the purified rEoGSTs1 protein, Glutathione S-transferase (GSH-ST) assay kit (Colorimetric method) (Cat. No. A004, Nanjing Jiancheng Bioengineering Institute, Nanjing, Jiangsu, China) was used, following the manufacturer's protocol. The pH of the reaction system was adjusted with PBS to 5.0, 6.0, 7.0, 8.0, and 9.0.

## Data processing

The *EoGSTs1* full-length cDNA was assembled from the DNA fragments using Seqman 7.1.0 software (DNAStar, Madison, WI, USA). Homology analysis of EoGSTS1 protein sequences in the NCBI database was performed using BLAST (*Johnson et al., 2008*). The physical and chemical properties of the EoGSTS1 protein were analyzed using software tools provided in the ExPASy-ProParam website (http://www.expasy.org), and cluster analysis was performed using MEGA 6.0 (*Tamura et al., 2013*). Clustal Omega (*Sievers et al., 2011*) was used in amino acid sequence alignment of EoGSTS1 and the GSTs of four other insects of order *Hemiptera*. The secondary structure of EoGSTS1 was predicted by CLC Genomics Workbench 11.0.1 (CLC Bio, Aarhus, Denmark), whereas the three-dimensional (3D) structure was predicted using I-TASSER software (*Roy, Kucukural & Zhang, 2010*). The functional domains were predicted based on the amino acid sequence deduced from the nucleotide sequence of the *EoGSTs1* gene by searching the conserved domain database (CDD) in the NCBI database (*Marchler-Bauer et al., 2006*). Phylogenetic reconstruction of the GSTs was performed using the maximum likelihood (ML) method and Bayesian inference (BI) with the assistance of iqtree 1.6.1 (*Nguyen et al., 2015*) and MrBayes 3.2 (*Ronquist et al., 2012*).

## RESULTS

### Detection results of total RNA

Both the 28S RNA and 18S RNA bands were observed as distinct bands (Fig. 1), suggesting that the extracted RNAs were of good quality. The OD260/280 ratio of the RNA was 1.92, indicating that the collected RNA was of high purity with no degradation.

### Amplification of the full-length cDNA

The 5′ end of the *EoGSTs1* cDNA fragment obtained by 5′ RACE was 223 bp in length. The middle fragment of the *EoGSTs1* amplified by intermediate fragment PCR was 480 bp in size. The 3′ end of the *EoGSTs1* cDNA obtained by 3′ RACE was 431 bp in length (Figs. 2A–2C). The full-length cDNA of *EoGSTs1* was an 841-bp sequence obtained through sequence assembly of the three DNA fragments; its coding region is 624-bp long, encoding a total of 207 amino acids, including the start codon ATG and the stop codon TGA (Dataset S1). The noncoding region at the 5′ end is 99 bp in length, and the 3′ end noncoding region is 118 bp in length followed by a 16-bp poly(A) tail (Fig. 3).

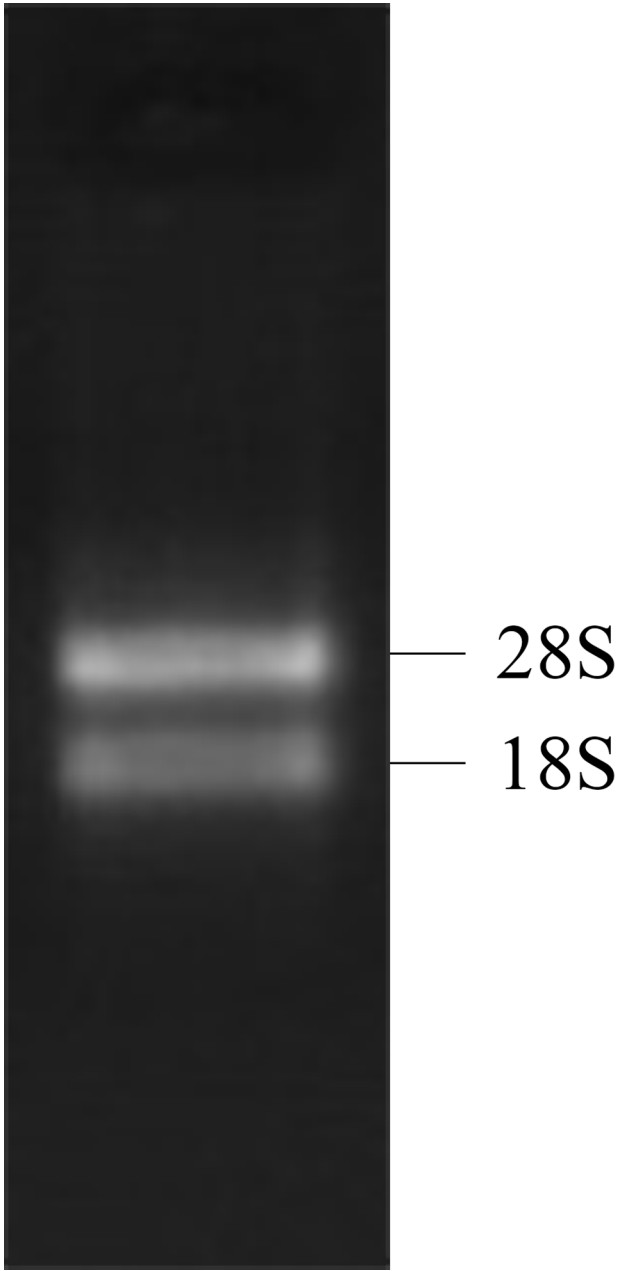

**Figure 1** 1% gel electrophoresis results of tatol RNA.

## Taxonomic identification of *EoGST*

The amino acid sequence of *EoGST* is highly homologous to GSTs in *Sub psaltriayangi* (AVC68800.1), *Locusta migratoria* (AHC08043.1), *Daktulosphaira vitifoliae* (AUN35388.1), and *Leptinotarsa decemlineata* (APX61045.1), with a similarity of 53%, 52%, 52%, and 50%, respectively.

The amino acid sequence of *EoGST* obtained in small green leafhoppers is shown in Table 2 along with 54 other GST sequences belonging to six GST families. Cluster analysis

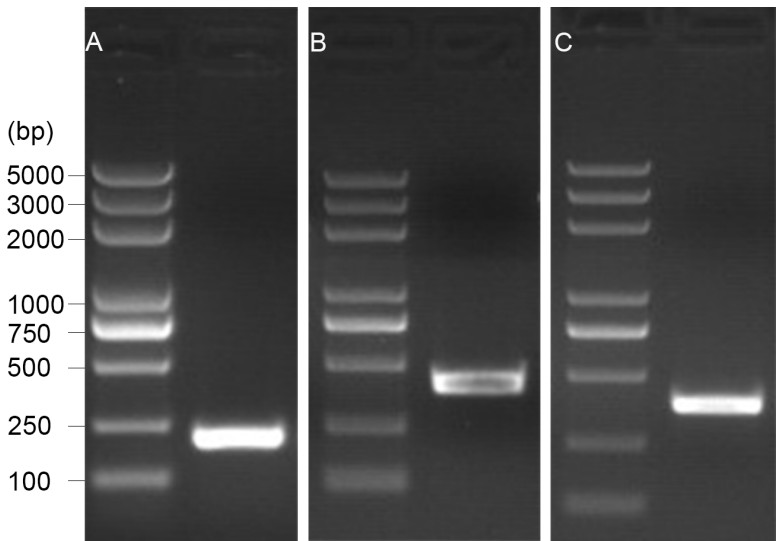

**Figure 2** **1.5% gel electrophoresis results of amplification products.** A, 5′ RACE PCR; B, middle fragment PCR; C, 3′ RACE PCR.

using the adjacency method demonstrated that the *EoGST* gene is clustered with the Sigma subfamily of GSTs (a high bootstrap value of 99%), indicating that *EoGST* is closely related to this particular subfamily (Fig. 4). However, it did not cluster with GSTs in the Delta or Epsilon subfamilies, indicating that it is distantly related to these two subfamilies. Based on these findings and mammalian GST gene nomenclature, the nucleotide sequence of the *E. onukii* GST (*EoGST, cluster-166.0*) was initially named as *EoGSTs1*, and the encoded protein sequence was designated as EoGSTS1.

## Bioinformatics analysis
### Physical and chemical properties of the EoGSTS1 protein

The molecular weight of EoGSTS1 is 23.68932 kDa, with a calculated isoelectric point (IEP) of 6.00. The number of negatively and positively charged residues is 30 and 29, respectively. Its N-terminus begins with methionine, and the half-life of the protein is 30 h. The instability index of the protein in solution is 31.87, the fat coefficient is 78.26, and the total average hydrophobicity index is −0.376. The protein contains a total of 3,331 atoms, including 1,078 carbon atoms, 1,662 hydrogen atoms, 276 nitrogen atoms, 305 oxygen atoms, and 10 sulfur atoms. It consists of 20 different types of amino acids, among which the top three are alanine (Ala, A), lysine (Lys, K), and glutamic acid (Glu, E), with respective counts of 19 (9.2%), 19 (9.2%), and 16 (7.7%). The rarely occurring amino acids included Cys (C) and His (H), with counts of 2 (1%) and 1 (0.5%), respectively (Table 3).

### EoGSTS1 protein sequence alignment

The catalytic domain of the protein contains 69 amino acid residues, as revealed in the SMART software analysis (*Letunic, Doerks & Bork, 2015*). Comparison of the protein sequences of EoGSTS1 and the GSTs of four other insects of order *Hemiptera* using Clustal Omega (Fig. 5) revealed the existence of multiple conserved domains.

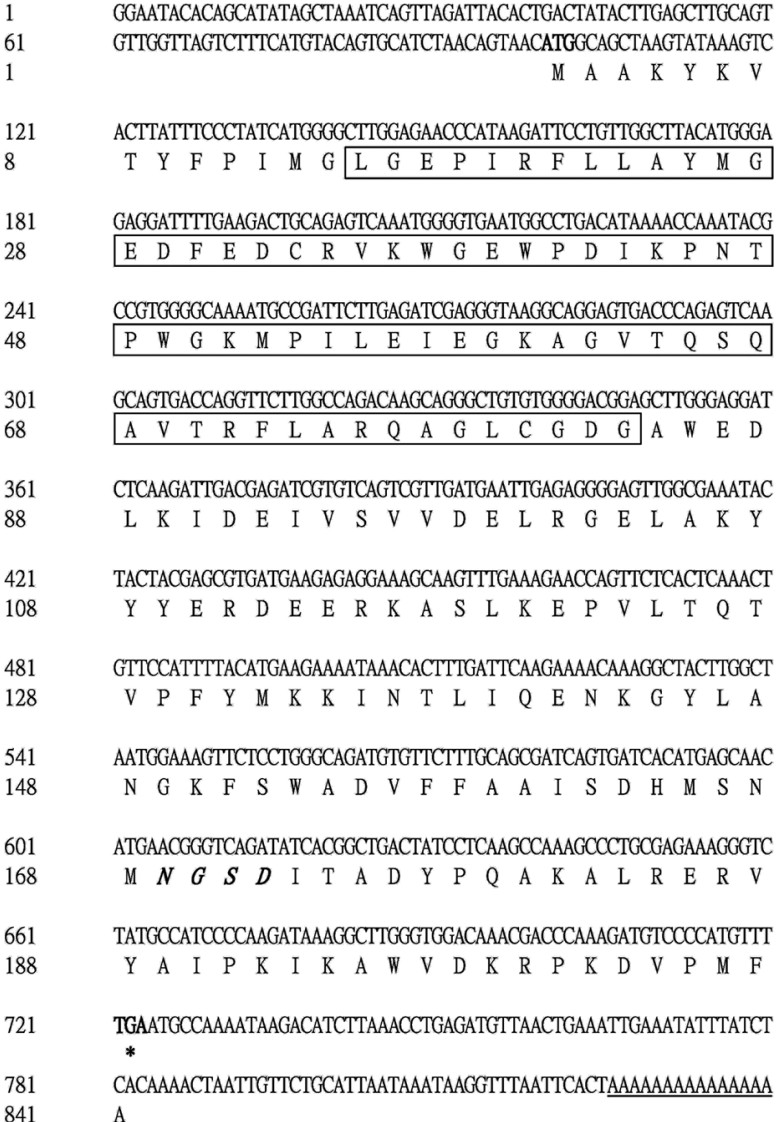

**Figure 3 Complete nucleotide sequence encoding *E. onukii* Matsuda and deduced amino acids of the cloned *EoGSTs1*.** Start codon (ATG), black bold; Putative polyadenylation signal, underline; Stop codon, black bold with an asterisk; Putative catalytic domain, rectangle; N-linked glycosylation sites, italic bold.

### Analysis of the EoGSTS1 protein structure

Based on its protein sequence, neither signal peptide nor transmembrane helix domains are present in the protein, but four potentially linear antigenic epitopes may exist in the protein sequence (amino acids 31–51, 81–88, 112–123, and 169–180).

The EoGSTS1 protein contains nine $\alpha$ helices, two $\beta$ sheets with a protein kinase C phosphorylation site, and one N-glycosylation site (Fig. 6). The protein belongs to the superfamily of GSTs and shares sequence similarities with GSTs in other insects (Fig. 7).

The protein sequence of EoGSTS1 contains two Asn-X-Ser/Thr motifs, one of which is an N-glycosylation site. The protein sequence also contains eight other potential

**Table 2 GST genes of different insect species and GenBank accession numbers.**

| Types | Species | Gene code | Accession number |
|---|---|---|---|
| Delta | *Tenebrio molitor* | TmGSTD | AIL23532.1 |
| | *Locusta migratoria* | LmGSTD | ADR30117.1 |
| | *Bombyx mori* | BmGSTD | BAD60789.1 |
| | *Culex pipiens* | CpGSTD | AEW07374.1 |
| | *Plutella xylostella* | PxGSTD | BAJ10978.1 |
| | *Lasioderma serricorne* | LsGSTD | AUO28661.1 |
| | *Chilo suppressalis* | CsGSTD | AKS40338.1 |
| | *Cnaphalocrocis medinalis* | CmGSTD | AIL29308.1 |
| | *Spodoptera litura* | SlGSTD | AIH07596.1 |
| Epsilon | *Drosophila willistoni* | DwGSTE | XP_002068808.1 |
| | *Spodoptera exigua* | SeGSTE | AHB18378.1 |
| | *Bombyx mori* | BmGSTE | NP_001108460.1 |
| | *Anopheles funestus* | AfGSTE | AHC31033.1 |
| | *Cnaphalocrocis medinalis* | CmGSTE | AIL29313.1 |
| | *Spodoptera litura* | SlGSTE | AIH07589.1 |
| | *Aedes aegypti* | AaGSTE | NP_001345908.1 |
| | *Bicyclus anynana* | BaGSTE | AON96571.1 |
| | *Anopheles cracens* | AcGSTE | ACY95463.1 |
| Theta | *Alitta succinea* | AsGSTT | ABQ82132.1 |
| | *Tenebrio molitor* | TmGSTT | AIL23552.1 |
| | *Mus musculus* | MmGSTT | CAA66666.1 |
| | *Carassius auratus* | CaGSTT | ABW96271.1 |
| | *Cyprinus carpio* | CcGSTT | BAS29977.1 |
| | *Channa punctata* | CpGSTT | ABY83769.1 |
| | *Procambarus clarkii* | PcGSTT | AXR98486.1 |
| | *Andrias davidianus* | AdGSTT | AYG85510.1 |
| | *Medicago truncatula* | MtGSTT | AET05000.1 |
| Omega | *Sus scrofa* | SsGSTO | AAF71994.2 |
| | *Halocynthia roretzi* | HrGSTO | BAD77935.1 |
| | *Schistosoma mansoni* | SmGSTO | AAO49385.1 |
| | *Tenebrio molitor* | TmGSTO | AIL23546.1 |
| | *Perna viridis* | PvGSTO | AGN03944.1 |
| | *Tigriopus japonicus* | TjGSTO | ACE81246.1 |
| | *Cyprinus carpio* | CcGSTO | BAS29980.1 |
| | *Kryptolebias marmoratus* | KmGSTO | AEM65182.1 |
| | *Spodoptera exigua* | SeGSTO | AHB18379.1 |
| | *Brassica napus* | BnGSTZ | AAO60042.1 |
| | *Arabidopsis thaliana* | AtGSTZ | AAO60039.1 |
| | *Tigriopus japonicus* | TjGSTZ | ACE81250.1 |
| | *Tenebrio molitor* | TmGSTZ | AIL23553.1 |

**Table 2** (*continued*)

| Types | Species | Gene code | Accession number |
|-------|---------|-----------|------------------|
| Zeta | *Panonychus citri* | PcGSTZ | AFD36889.1 |
| | *Azumapecten farreri* | AfGSTZ | ADD82544.1 |
| | *Vibrio sinaloensis* | VsGSTZ | EGA70514.1 |
| | *Tetrahymena thermophila* | TtGSTZ | EAR89084.1 |
| | *Cyprinus carpio* | CcGSTZ | BAS29981.1 |
| | *Antheraea pernyi* | ApGSTS | ADC32118.1 |
| | *Bombyx mori* | BmGSTS | BAD91107.1 |
| | *Tenebrio molitor* | TmGSTS | AIL23551.1 |
| | *Chilo suppressalis* | CsGSTS | ADD14027.1 |
| Sigma | *Daphnia magna* | DmGSTS | AOQ25845.1 |
| | *Lasioderma serricorne* | LsGSTS | AUO28662.1 |
| | *Operophtera brumata* | ObGSTS | KOB58098.1 |
| | *Mytilus galloprovincialis* | MgGSTS | AFQ35985.1 |
| | *Cnaphalocrocis medinalis* | CmGSTS | AIZ46905.1 |

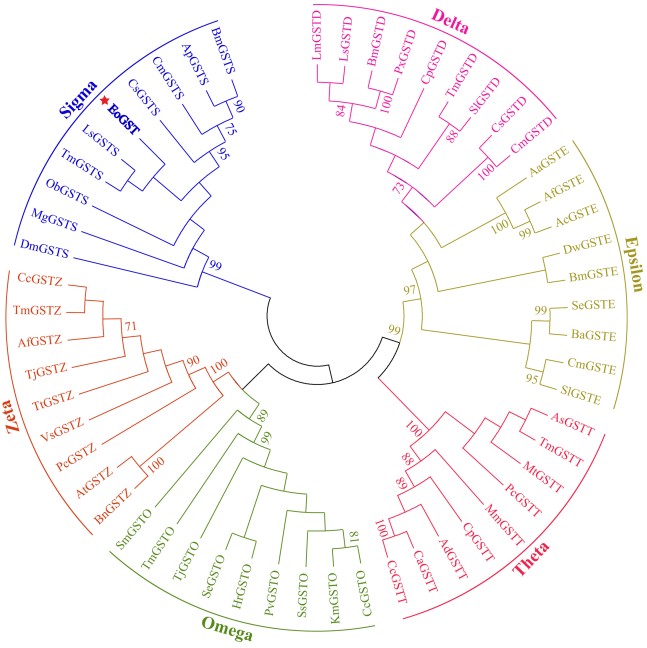

**Figure 4** **Clustering analysis of GST genes of different insect species based on the neighbouring method.** Numbers on the branch represent the bootstrap value (bootstrap >70%).

phosphorylation sites: four serine phosphorylation sites located at positions 95, 118, 152, and 166 in the peptide chain; two threonine phosphorylation sites located at positions 47 and 125 in the peptide chain; and two tyrosine phosphorylation sites located at positions 148 and 188 in the peptide chain (Table 4).

The 3D structure of the EoGSTS1 protein was predicted using the *Blattella germanica* GST (BgGST) protein (PDB ID: 4Q5R) as reference (as shown in Fig. 8, the C-terminus is

**Table 3** The chemical composition of the cloned fragment of *EoGSTs1*.

| Amino acid | Number | Proportion of quantity | Mass ratio | Amino acid | Number | Proportion of quantity | Mass ratio |
|---|---|---|---|---|---|---|---|
| Ala (A) | 19 | 9.20% | 6.18% | Lys (K) | 19 | 9.20% | 10.14% |
| Arg (R) | 10 | 4.80% | 6.36% | Met (M) | 8 | 3.90% | 4.36% |
| Asn (N) | 6 | 2.90% | 2.89% | Phe (F) | 9 | 4.30% | 5.43% |
| Asp (D) | 14 | 6.80% | 6.80% | Pro (P) | 12 | 5.80% | 5.04% |
| Cys (C) | 2 | 1.00% | 0.88% | Ser (S) | 7 | 3.40% | 2.68% |
| Gln (Q) | 6 | 2.90% | 3.20% | Thr (T) | 8 | 3.90% | 3.48% |
| Glu (E) | 16 | 7.70% | 8.59% | Trp (W) | 6 | 2.90% | 4.47% |
| Gly (G) | 14 | 6.80% | 3.84% | Tyr (Y) | 10 | 4.80% | 6.61% |
| His (H) | 1 | 0.50% | 0.57% | Val (V) | 13 | 6.30% | 5.56% |
| Ile (I) | 13 | 6.30% | 6.22% | Pyl (O) | 0 | 0.00% | 0.00% |
| Leu (L) | 14 | 6.80% | 6.70% | Sec (U) | 0 | 0.00% | 0.00% |

```
EoGST    MAAKYKVTYFPIMGLGEPIRFLLAYMGEDFEDCRVKWGEWPDIKPNTPWGKMPILEIEGK
SyGST    MALKYKLTYFDGKGLAEPIRYILSYMGEEFEDDRFTKEEWPLIKPSTPFGKAPVLSVDGK
SfGST    -MSTYKLTYFPVTALGEPIRWLMSYLDIKFEDYRFEREQWPSIKPTTPFGQVPVLEIDGK
LsGST    -MSAYKLTYFPVTALGEPIRWLLSYLDIKFEDYRFEREQWPSIKPTTPFGQVPVLEIDGK
NlGST    -MSGYKLTYFPVTALGEPIRWMMSYLDIKFEDYRFEREQWPSIKPTTPFGQVPVLEIDGK
           **:***    .*.****:::*:.  *** *.    :** ***.**:*: *:*..:**

EoGST    AGVTQSQAVTRFLARQAGLCGDGAWEDLKIDEIVSVVDELRGELAKYYYERDEERKASLK
SyGST    Q-LCQSVALTRYLAKKADLVGKDEWEDLHIDMIVDTIGDLRQAIASYYYDPDEESRAAKK
SfGST    A-VWQSVAISRYFGKKADLAGKDEWESLMIDVIVDTFSDFRLAVGKWFYESDEATKKNLE
LsGST    V-VWQSVAISRYFGKKADLAGKDEWEALMIDVIVDTFTDFRMAVGKWFYESDEAAKKKLE
NlGST    S-VWQSVAISRYFGKKADLAGKDEWESLMIDVIVDTFTDFRLAVGKWFYESDEATKKKLE
           : ** *::*:::.::*.* *.. ** * ** **... ::*   :...::*: **  :    :

EoGST    EPVLTQTVPFYMKKINTLIQENKGYLANGKFSWADVFFAAISDHMSNMNGSDITADYPQA
SyGST    EPLLNETIPFYMSKFENIANENNGYLANGRLSWADIYLVALSEYMSSIAGTDLLEPYPTL
SfGST    KPLFETTIPFYLEKFDSKIKENGGFLANGKLSWGDIYFVAVSGYVNHMLGFNMSEKYDNI
LsGST    IPLFETTVPFYLEKFDSTIKENGGFLANGKLSWGDIFFVAVSGYVNHMLGFNMSDKYENI
NlGST    KPLLETTVPFYLEKFDSTIKENGGFLANGKLSWGDIYFVATSGYINHMLGFNMSDKYENI
           *::   *:***:.*::.   :** *:****.:**.*:::.* * ::. : * ::    *

EoGST    KALRERVYAIPKIKAWVDKRPKDVPMF
SyGST    TSLKEIVWGIPKIKEWIEKRPKTDI--
SfGST    KALCEKVSAIPKIKEWIAKCPAGI---
LsGST    KALCEKVSAIPKIKEWIDKRPAGI---
NlGST    KALCEKVAAIPKIKEWIDKRPAGI---
           .:* * *  .***** *: * *
```

**Figure 5** Comparison of GST amino acid sequence of some insects (Hemiptera). '*' represents identical amino acid residues, '-' represents sequence deletion of RNA at this position, '.' and ':' represent conservatism of RNA sequence at different degrees, no identifier represents the sequence does not have conservation.

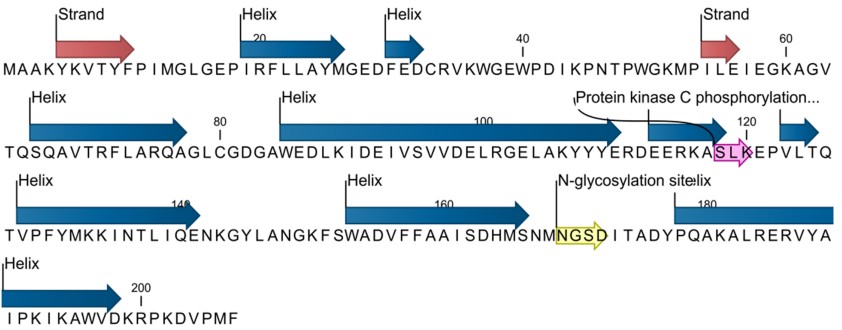

**Figure 6** Predicted secondary structure and annotated functional sites of the *EoGSTs1* protein. The protein kinase C phosphorylation site is shown in purple and the N-glycosylation site is shown in yellow.

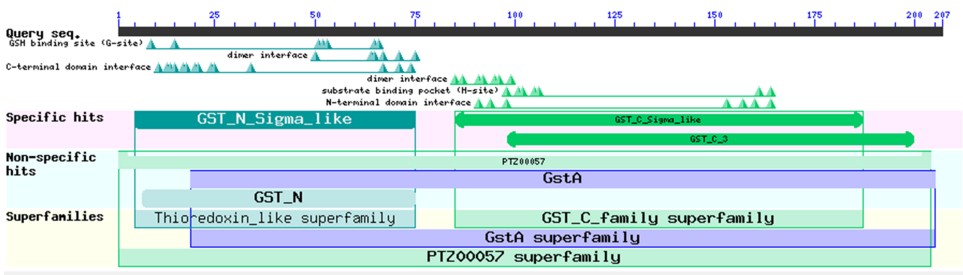

**Figure 7** Prediction of the functional domains of the *EoGSTs1* protein. Small triangle: conservative character or amino acid of the conserved sites, specific hits: specific hits domains, non-specific hits: non-specific hits domains, superfamilies: superfamilies that corresponds to the above-mentioned matching domains.

**Table 4  Predicted phosphorylation sites.**

| Position | Sequence | Score | Classification | Position | Sequence | Score | Classification |
|---|---|---|---|---|---|---|---|
| 95 | DEIVSVVDE | 0.986 | *S* | 47 | IKPNTPWGK | 0.801 | *T* |
| 118 | ERKASLKEP | 0.998 | *S* | 125 | EPVLTQTVP | 0.597 | *T* |
| 152 | NGKFSWADV | 0.985 | *S* | 145 | ENKGYLANG | 0.932 | *Y* |
| 166 | SDHMSNMNG | 0.546 | *S* | 188 | RERVYAIPK | 0.890 | *Y* |

denoted in red and the N-terminus in blue). The C-terminus is composed of 207 amino acids, and the N-terminus contains 204 amino acids. The C-score of the protein structural comparison is 1.21, indicating that it is highly similar to 4Q5R in terms of protein folding and secondary structure. The TM-score, which represents the structural similarity between the target sequence (EoGSTS1) and the template protein sequence (BgGST), is $0.88 \pm 0.07$.

## Phylogenetic analysis

The phylogenetic tree of the full-length EoGSTS1 protein sequence and GST sequences of insects from five other orders, including *Hemiptera, Orthoptera, Coleoptera, Odonata,* and *Hymenoptera* (Table 5), were constructed using BI phylogenetic analysis and an ML-based

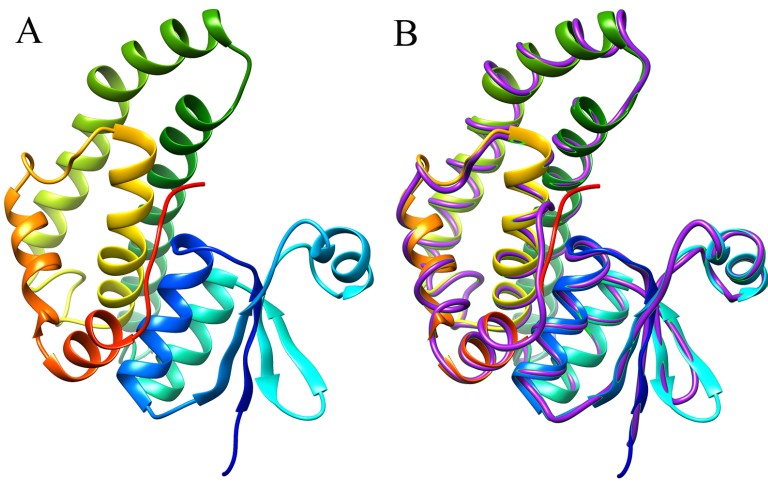

**Figure 8** **Predicted 3D structure and main chain interface structure of the *EoGSTs1* protein.** (A) Predicted 3D structure of the *EoGSTs1* protein of *E. onukii* Matsuda. (B) Superimposed prediction model and native cartoon structures of *E. onukii* Matsuda *EoGSTs1* and *Blattella germanica* GST. Rainbow structure represents *E. onukii* Matsuda *EoGSTs1*. Purple line represents the alpha carbon backbone trace of *B. germanica* GST.

method. The topological structures of the phylogenetic trees generated using either ML or BI displayed substantial similarities (Fig. 9). Cluster analysis indicated that the EoGSTS1 is clustered with known insect GSTs of order *Hemiptera* but not with GSTs from insects of order *Odonata* or *Hymenoptera*. These results suggest that *E. onukii* Matsuda is most closely related to insects of order *Hemiptera* and relatively closely related to those in order *Orthoptera* but distantly related to insects of orders *Hymenoptera, Coleoptera,* and *Odonata.*

## Expression of the rEoGSTs1 protein

The recombinant expression plasmid pCzn1-*EoGSTs1*-target was transformed into *E. coli.* Arctic Express and the pCzn1/*EoGSTs1* fusion protein was successfully expressed with the induction of IPTG. SDS-PAGE analysis showed that rEoGSTs1 was expressed in the supernatant and mainly occurred in a soluble form (Fig. 10A). Western blotting revealed a specific band in the corresponding position, which indicated that this band is the target band (Fig. 10B). The concentration of the target protein that was determined by the BCA method was 1.2 mg/mL.

## Enzymatic analysis of EoGSTS1

The kinetic data of EoGSTS1 showed that the rEoGSTs1 protein could catalyze CDNB with Km of $0.21 \pm 0.06$ mmol/L and Vmax of $14.02 \pm 1.40$ μmol/min·mg (Fig. 11A). At different temperatures, the specific activity of EoGSTS1 initially gradually increased, and then decreased after reaching the highest value at 25 °C, and reached the lowest value at 60 °C. The lowest enzyme activity was about 4.13% of the highest one (Fig. 11B). At different pH levels, the enzyme activity of EoGSTS1 initially increased and then decreased
**Table 5  GST genes of different insect species and GenBank accession numbers.**

| Species | Gene code | Accession number |
|---|---|---|
| *Sogatella furcifera* | SyGST | AFJ75815.1 |
| *Halyomorpha halys* | HhGST | XP_014283901.1 |
| *Laodelphax striatella* | LsGST | AEY80032.1 |
| *Nilaparvata lugens* | NlGST | XP_022201420.1 |
| *Riptortus pedestris* | RpGST | BAN21228.1 |
| *Cimex lectularius* | ClGST | XP_024082892.1 |
| *Sogatella furcifera* | SfGST | AFJ75815.1 |
| *Locusta migratoria* | LmGST | AHC08043.1 |
| *Schistocerca gregaria* | SgGST | AEV89756.1 |
| *Leptinotarsa decemlineata* | LdGST | APX61045.1 |
| *Tribolium castaneum* | TcGST | XP_967475.1 |
| *Anoplophora glabripennis* | AgGST | XP_018560985.1 |
| *Tenebrio molitor* | TmGST | AIL23548.1 |
| *Aethina tumida* | AtGST | XP_019878993.1 |
| *Agrilus planipennis* | ApGST | XP_018334284.1 |
| *Blattella germanica* | BgGST | PSN56155.1 |
| *Zootermopsis nevadensis* | ZnGST | XP_021913534.1 |
| *Periplaneta americana* | PaGST | AVA17428.1 |
| *Apis cerana cerana* | AcGST | PBC25817.1 |
| *Athalia rosae* | ArGST | XP_012268124.1 |
| *Copidosoma floridanum* | CfGST | XP_014210579.1 |
| *Apis florea* | AfGST | XP_003694329.2 |
| *Apis mellifera* | AmGST | XP_026295805.1 |
| *Megachile rotundata* | MrGST | XP_003703954.1 |
| *Microplitis demolitor* | MdGST | XP_008550325.1 |

with increasing pH, peaked at pH 7, and at pH 5, the enzyme activity was 73.67% of that at pH 7 (Fig. 11C).

## DISCUSSION

GSTs represent a superfamily of genes that are widely present in many organisms and play critical roles in the resistance to foreign substances in insects. In this study, we used transcriptome analysis to screen overexpressed genes induced by thiamethoxam at the early stage of induction. Based on the results, the cDNA of *EoGSTs1 cluster-166.0* was selected, and its full-length cDNA was amplified by qPCR. The end products showed that the full-length cDNA sequence of *EoGSTs1* was 841 bp, with a 624-bp coding region that is predicted to generate a protein consisting of 207 amino acids.

The results from the homology search and cluster analysis suggest that the EoGSTS1 protein is highly similar to *Sub psaltriayangi* (AVC68800.1) GST and *Locusta migratoria* GST (AHC08043.1) and clustered with the GSTs in the Sigma subfamily with stronger support (bootsrap value > 90%). Thus, it is believed to belong to the Sigma GST gene subfamily, which coincides with previous findings that the Sigma subfamily of GSTs is

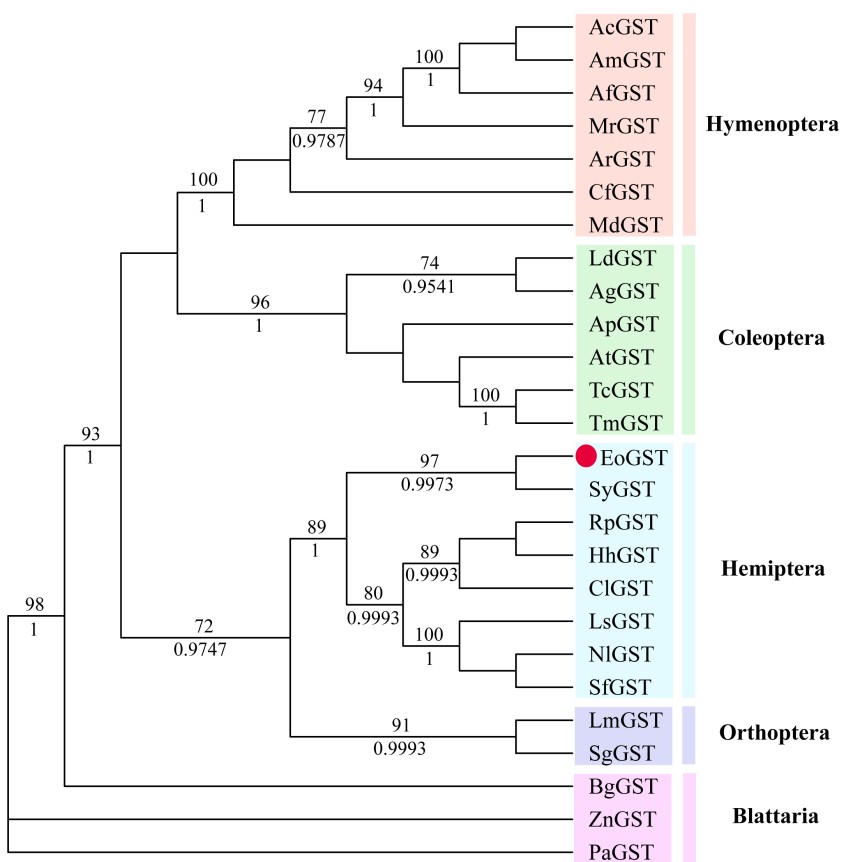

**Figure 9** The phylogenetic tree was constructed with ML (best model: LG + I + G4) and BI (best model: LG + I + G, −ln L = −7163.4499) methods based on the GST genes of different insects. Numbers on the branch indicate the ML bootstrap values (>70%), whereas Bayesian posterior possibilities (>90%) are shown under the branch.

involved in chemical resistance in other insects. For example, *Zhou et al. (2012)* reported that the upregulation of *LsGSTs2* gene expression (Sigma subfamily) can be induced by cypermethrin, chlorpyrifos, and fipronil in *Laodelphax striatellus*. The results from the gene expression study of GST by *You et al. (2015)* showed that *PxGSTs1* (another gene in the Sigma subfamily) was significantly upregulated in the resistant strain of *Plutella xylostella*. In the present study, the expression of *EoGSTs1* in the chemically induced group of small green leafhoppers was 6.6538 times that of the untreated group; therefore, we believe that *EoGSTs1* plays an important role in resistance to thiamethoxam in small green leafhoppers.

The instability index of the EoGSTS1 protein in solution is below the threshold of 40, and the average hydrophobicity is a negative value, indicating that the protein is a stable hydrophilic protein (*Nie, Wu & Zhang, 2006*; *Wanyonyi et al., 2011*). Using SignalP 4.1 and subcellular localization analysis, we found that the protein lacks sequences that direct subcellular localization such as signal peptides, lysosomes, peroxidase, or mitochondria. There are no transmembrane domains in EoGSTS1. Therefore, we conclude that EoGSTS1

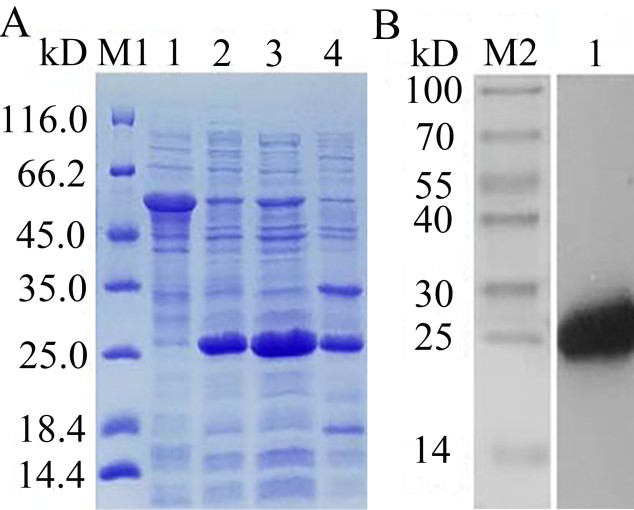

**Figure 10** SDS-PAGE analysis of expression (A) and Western Blot analysis of fusion protein purification (B). (A) M1: Protein molecular weight marker; 1: Expressed product of rEoGSTs1 protein without induction with IPTG at 11 °C; 2: Expressed product of rEoGSTs1 protein with induction with IPTG at 11 °C; 3: The supernatant of rEoGSTs1 protein with induction with IPTG at 11 °C; 4: The precipitant of rEoGSTs1 protein with induction with IPTG at 11 °C. (B) M2: Protein molecular weight marker; 1: Purified sample.

is a nontransmembrane GST protein located in the cytoplasm with a single functional structure (*Li et al., 2018a*; *Li et al., 2018b*).

Functional domain prediction based on the protein structure of EoGSTS1 showed that it contains a GST conserved domain and belongs to the protein superfamily. It consists of an N-terminal domain starting with a methionine and a C-terminal domain with multiple $\alpha$ helices and $\beta$ sheets dispersing throughout the structure. These motifs play an important role in linking polypeptides together and forming the protein structure (*Sun, 1994*). Multiple sites for potential phosphorylation also exist in the protein sequence, which can be divided into three types, with serine phosphorylation as the major one. Phosphorylation site predictions can provide insights into the important biological processes occurring in *E. onukii* such as cell growth and development, signal transduction, and gene expression. Proteins participate in various essential biological activities in organisms, including signal transduction, enzyme catalysis, and cellular transportation (*Wu, 2010*). Therefore, the 3D structural information of a protein can be very valuable for its related biological or medical studies such as configuring the structural and functional composition of the protein and designing protein-targeted drug molecules (*Kuhlman et al., 2003*). In the present study, we used the threading method to predict the 3D structure of EoGSTS1. The results demonstrated that EoGSTS1 in 3D is highly similar to that of the 4Q5R protein from the German cockroach *Blattella germanica*. The C-score of the compared proteins is $>-5$ and $<2$, and the TM-score is $>0.5$ (*Yang & Zhang, 2015*; *Xu & Zhang, 2010*), indicating that the prediction model is sound with correct topology and high reliability in terms of quality assessment.

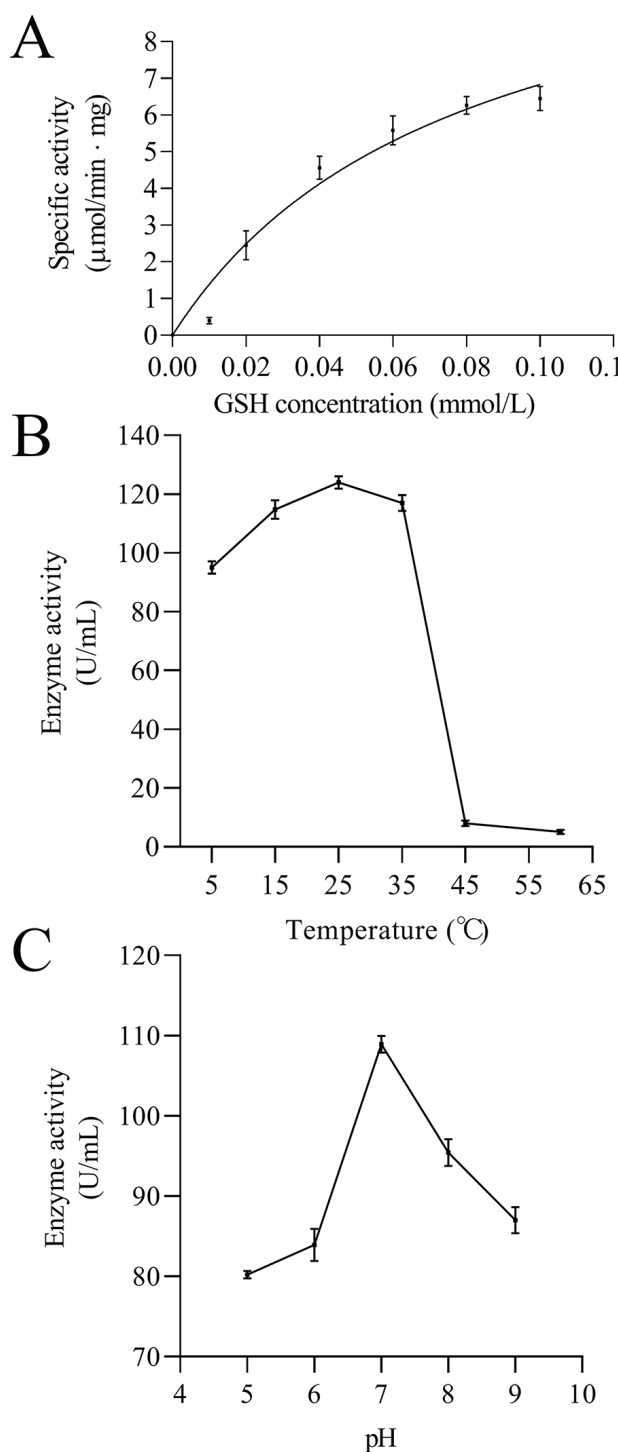

**Figure 11 Enzymatic kinetics of the purified rEoGSTs1 protein (A) and influence of temperature (B) and pH (C) on the enzyme activity of the purified rEoGSTs1 protein.** Data in the figure are mean ± SE.

Using BLAST alignment, we established that the amino acid sequence encoded by *EoGSTs1* is highly similar to that of *Sub psaltriayangi,* but its similarities with insects belonging to orders *Orthoptera, Coleoptera, Odonata,* and *Hymenoptera* did not show notable differences, suggesting that GSTs are not highly conserved across different orders. The EoGSTS1 clusters with a known insect GST in order *Hemiptera* in ML-based and BI phylogenetic trees; however, within that gene cluster, EoGSTS1 forms another branch with *Cicadidae* GST, indicating a close genetic association between the two. However, its genetic relationships with GSTs in insects that are not in order *Hemiptera* appear to be distantly related, as it is completely separated in the phylogenetic tree. This result further verifies the close genetic association between EoGSTS1 and the known GST gene in order *Hemiptera* and the conservation of the GST genes within order *Hemiptera.*

The target protein was purified by Ni affinity chromatography, the results of SDS-PAGE detection and western blot detection coincided with prediction results of the molecular weight of protein (23.68932 kDa), indicating that the expression product had not been extensively processed and modified. The determination of enzyme activity showed that the specific activity of rEoGSTs1 protein reached the highest value at 25 °C and a pH of 7, which is similar to the results of glutathione transferase in silkworm (*Yamamoto et al., 2006*). CNDB is a kind of insoluble and poisonous chemical compound that has high reactivity with many GST isoforms; therefore, it is often used as a model substrate to detect whether GSTs have detoxification activity (*Tan et al., 2014*; *Deponte, 2013*). In this study, the kinetic parameters of EoGSTS1 showed that it has catalytic activity on the model substrate CDNB, and compared to the results of the activity of BmGSTD in silkworm (*Yamamoto et al., 2012*), Vmax of the EoGSTS1 protein is higher than that of BmGSTD, which indicates that rEoGSTs1 protein has good detoxification function. Together with the differential gene profiling results, we propose that EoGSTS1 plays an important role in the detoxification of *E. onukii* to thiamethoxam.

## CONCLUSIONS

In this study, the full-length cDNA of the *E. onukii* GST gene *EoGSTs1* was successfully amplified by qPCR. EoGSTS1 belongs to the Sigma subfamily of GSTs and is a stable hydrophilic protein located in the cytoplasm and a nontransmembrane GST protein with a single functional structure. The EoGSTS1 protein has multiple motifs of $\alpha$ helices and $\beta$ sheets distributed throughout its N- and C-termini. The protein sequence indicates the existence of multiple potential phosphorylation sites, with serine phosphorylation sites as the major type. The 3D structure of the EoGSTS1 protein is highly similar to that of 4Q5R, a GST protein in the German cockroach *B. germanica*. EoGSTS1 is closely related to insect GSTs of order *Hemiptera* but is distantly related to GSTs of insects of order *Blattodea.* The results of recombinant expression and purification of EoGSTS1 *in vitro* and the enzyme activity assay of purified protein show that EoGSTS1 possesses detoxification activity.

### Funding

This work was supported by the Program of Excellent Innovation Talents, Guizhou Province (No. 20154021), the Program of Science and Technology Innovation Talents Team, Guizhou Province (No. 20144001), the Key Agricultural Science and Technology Projects in Guizhou Province (NY [2010] 3026) and the International Cooperation Base for Insect Evolutionary Biology and Pest Control (No. 20165802). The funders had no role in study design, data collection and analysis, decision to publish, or preparation of the manuscript.

### Grant Disclosures

The following grant information was disclosed by the authors:
Program of Excellent Innovation Talents, Guizhou Province: 20154021.
Program of Science and Technology Innovation Talents Team, Guizhou Province: 20144001.
Key Agricultural Science and Technology Projects in Guizhou Province: NY [2010] 3026.
International Cooperation Base for Insect Evolutionary Biology and Pest Control: 20165802.

### Competing Interests

The authors declare there are no competing interests.

### Author Contributions

- Yujie Zhang conceived and designed the experiments, performed the experiments, analyzed the data, contributed reagents/materials/analysis tools, prepared figures and/or tables, authored or reviewed drafts of the paper.
- Wenlong Chen authored or reviewed drafts of the paper, provided Research Fund Projects.
- Ming Li authored or reviewed drafts of the paper, provided the guiding suggestions for experiment.
- Lin Yang authored or reviewed drafts of the paper, provided the guiding suggestions for revision.
- Xiangsheng Chen conceived and designed the experiments, contributed reagents/-materials/analysis tools, authored or reviewed drafts of the paper, approved the final draft.

### DNA Deposition

The following information was supplied regarding the deposition of DNA sequences:
The *EoGSTs1* sequence is available at GenBank: MK443501.

### Data Availability

Raw data are available in Table 1, Table 2, Table 5 and Datasets S1–S4.

## Supplemental Information

Supplemental information for this article can be found online at http://dx.doi.org/10.7717/peerj.7641#supplemental-information.

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
