# Peer review of "Cloning, phylogenetic research, and prokaryotic expression study of the metabolic detoxification gene EoGSTs1 in Empoasca onukii Matsuda"

_PeerJ, doi:10.7717/peerj.7641_

## Round 0.1 · original submission · Major Revisions

There are several questions raised by the reviewers. Please address all of them, specifically about the novelty of this work by the Reviewer 2. Unsatisfactory answer or insufficient work may lead to rejection of this manuscript.

Reviewer 1 ·

Basic reporting

The manuscript titled “Cloning, sequence analysis and phylogenetic research study of the metabolic detoxification gene, EoGSTs1, in Empoasca onukii Matsuda” by Zhang et al. reports the characterization of glutathione S-transferase from Empoasca onukii. The study is providing insights for the future use of EoGST protein for the management of chemical resistance in leafhoppers. The manuscript is overall well-structured and focused. The findings are interesting and will be utilized by the scientific community. However most of the conclusions are presented based on the bioinformatic evidences only.

Experimental design

Introduction, materials and methods and discussion section is well written and is clearly providing all the necessary details. The experiments are well executed.

Validity of the findings

I have only minor concerns,

Result section in few places is containing the part of the method that should go with materials and methods section, for e.g., Line 128, Please move all the methods part in result under methods section.

Please comment on the activity of EoGST. Authors should consider doing activity assay of EoGST in vitro.

Please elaborate the figure legends. Mainly figure 1, 5 and 7.

Additional comments

Manuscript is well structured, focused and experimentally sound and can be considered for publication in PeerJ after minor revision.

Reviewer 2 ·

Basic reporting

In this manuscript entitled “Cloning, sequence analysis and phylogenetic research study of the metabolic detoxification gene, EoGSTs1, in Empoasca onukii Matsuda” authors cloned full-length cDNA of EoGSTs1 from Empoasca onukii Matsuda and used several in silico tools to perform some analysis to reveal characteristics. Manuscript is short but well written. This is not the first instance of cloning GST from Empoasca onukii Matsuda. An 823 bp of GST cDNA was already been reported in PMID: 30296272. Although authors made effort to clone EoGSTs1 and for some in-silico analysis of the GST protein, this manuscript lacks novelty and biological validation by sufficient experiments to publish in “PeerJ”.

Experimental design

No comment

Validity of the findings

No comment

Additional comments

1) This is not the first instance of cloning GST from Empoasca onukii Matsuda. An 823 bp of GST cDNA was already been reported in PMID: 30296272.
2) In Line 81-82, authors mentioned that they performed screening of transcriptome data to identify differentially expressed genes upon chemical treatment. But authors didn’t show any analysis in the manuscript. Need explanation.
3) Why authors used specific concentration of thiamethoxam and time point. They could have showed some kinetics of expression of GST at least.
4) Did authors submit the sequence in public database?
5) Authors could have purified and characterized the protein which would makes manuscript acceptable. some in vitro detoxification studies can also be done?
6) Line 66, Expand DEM
7) Ref missing in line 172.

---

## Round 0.2 · Minor Revisions

Please go through the remaining comments suggested by both the reviewers.

Reviewer 1 ·

Basic reporting

The revised manuscript titled “Cloning, sequence analysis and phylogenetic research study of the metabolic detoxification gene, EoGSTs1, in Empoasca onukii Matsuda” by Zhang et al has reported most of my concerns. However, I have still found many grammatical errors, spelling mistakes and ambiguous sentences that need to be corrected (especially in the method section). Therefore, the manuscript needs serious attention in the writing part, before being considered for the publication. Authors could consider taking help from any native english speaker for the language correction in the manuscript.

Experimental design

Experiments are well executed. Please consider reducing the number of figures by combining figures and as different parts of same figure wherever possible, it would help the future readers of the manuscript to understand it better and will make the manuscript aesthetically sound.
Some more comments:
Line 135: Correct the name of restriction enzyme.
Line 136: Replace clone strain with cloning strain
Line 132: Revise the method section as it is poorly written and contains number of grammatical errors and typos.
Line 318: Remove “an”

Validity of the findings

Findings are interesting.

Additional comments

Manuscript is experimentally sound and can be considered for publication in PeerJ after minor revision.
Please check for spellings and typos present throughout the manuscript, even the rebuttal letter contains number of spelling mistakes.

Reviewer 2 ·

Basic reporting

I appreciate the author's efforts for corrections made in the manuscript. They answered most of my concerns. But I have the following concerns which need to be addressed before acceptance.

Experimental design

no comment

Validity of the findings

no comment

Additional comments

Comments to authors
- English language need to have relook for newly added paragraphs to the draft and correct where ever required
- Line 44, change enzymes assay to enzyme activity assay
- Remove common in line 45
- Mention at first instance recombinant EoGSRs1 (rEoGSTs1) and Use it as a synonym throughout the manuscript where ever your mentioning recombinant protein or recombinant enzyme.
- Use of unique synonym EoGST vs EoGSTs1?? Throughout the manuscript
- Line 122, correct -3 ul
- Mention Top10 strain correctly. Ecoli top10??
- Line 180/181, use term primary antibody instead of first antibody, secondary HRP antibody instead second antibody
- Line 319, Protein gel?? Means SDS-PAGE?

---

## Round 0.3 · accepted · Accept

Authors have incorporated all the suggested changes.

Reviewer 1 ·

Basic reporting

The manuscript by Zhang et al. has improved a lot after revision and can now be considered for publication in PeerJ.

Experimental design

no comments

Validity of the findings

no comments

Additional comments

Authors have done nice job in revising the manuscript.

Reviewer 2 ·

Basic reporting

The manuscript was significantly improved and now suitable to publish in PeerJ. I would recommend accepting this article.

Experimental design

No comment

Validity of the findings

No comment

Additional comments

The manuscript was significantly improved and now suitable to publish in PeerJ. I would recommend accepting this article.

Minor:
make a space between the and multiscan in the line 179
In line 162, authors wrote "conducted PVDF transmembrane". Please correct. IS that PVDF transfter?